# MATI: Multimodal Adaptive Tracking Integrator for Robust Visual Object Tracking

**DOI:** 10.3390/s24154911

**Published:** 2024-07-29

**Authors:** Kai Li, Lihua Cai, Guangjian He, Xun Gong

**Affiliations:** 1Changchun Institute of Optics, Fine Mechanics and Physics, Chinese Academy of Sciences, Changchun 130033, China; likai221@mails.ucas.ac.cn (K.L.); heguangjian@ciomp.ac.cn (G.H.); gongxun@ciomp.ac.cn (X.G.); 2University of Chinese Academy of Sciences, Beijing 100049, China

**Keywords:** visual object tracking, vision transformer, adaptive tracking mechanism, multimodal dataset

## Abstract

Visual object tracking, pivotal for applications like earth observation and environmental monitoring, encounters challenges under adverse conditions such as low light and complex backgrounds. Traditional tracking technologies often falter, especially when tracking dynamic objects like aircraft amidst rapid movements and environmental disturbances. This study introduces an innovative adaptive multimodal image object-tracking model that harnesses the capabilities of multispectral image sensors, combining infrared and visible light imagery to significantly enhance tracking accuracy and robustness. By employing the advanced vision transformer architecture and integrating token spatial filtering (TSF) and crossmodal compensation (CMC), our model dynamically adjusts to diverse tracking scenarios. Comprehensive experiments conducted on a private dataset and various public datasets demonstrate the model’s superior performance under extreme conditions, affirming its adaptability to rapid environmental changes and sensor limitations. This research not only advances visual tracking technology but also offers extensive insights into multisource image fusion and adaptive tracking strategies, establishing a robust foundation for future enhancements in sensor-based tracking systems.

## 1. Introduction

Visual object-tracking technology is critical for applications requiring precise real-time responses, such as earth observation, environmental monitoring, and military reconnaissance. Despite substantial advancements in tracking algorithms under typical conditions, maintaining accuracy and stability in extreme environments—characterized by target occlusion, low-light settings, and complex backgrounds—presents significant challenges [1]. These difficulties are exacerbated when tracking dynamic targets like aircraft, which are subject to rapid movements, appearance changes, and environmental disturbances from atmospheric interference, biological activity, and ground clutter. Additionally, the performance of traditional visible light sensors often declines under extreme or variable lighting conditions, while infrared sensors, despite their usefulness in detecting thermal variations, encounter limitations due to inherent resolution constraints and coarse textures. These issues become particularly acute during thermal crossover events, where reliable tracking and differentiation of targets from their backgrounds are critically impeded.

To overcome existing challenges and enhance the robustness of tracking systems, initial efforts predominantly relied on traditional filtering algorithms [2,3,4,5,6], which laid the groundwork for understanding and developing tracking systems. As technology advanced, many researchers [7,8,9,10,11,12,13,14] transitioned to deep learning approaches, utilizing CNNs or Transformers to develop more efficient single-modality target tracking algorithms. Building on this technological progression, several studies [15,16,17,18,19,20,21,22,23,24] have proposed the fusion of visible and infrared images, capitalizing on the complementary advantages of both modalities. This study introduces an innovative adaptive multimodal image object-tracking model that further refines these approaches by dynamically adjusting to changing environmental conditions and the availability of different modal types. Unlike most existing methods [25,26,27,28,29,30,31], which also leverage fusion strategies to enhance target tracking capabilities, our approach uniquely considers the practical engineering constraints, accommodating scenarios where certain image sensors may not need to be activated or cannot be used effectively. A key innovation of the MATI model is its dynamic adjustment capability according to the image source, allowing it to maintain high processing efficiency across diverse environmental conditions. Notably, the crossmodal compensation (CMC) module plays a pivotal role in extracting and integrating features from different modalities, while the token spatial filtering (TSF) module enhances attention distribution by dynamically adjusting the retention ratio of feature vectors, thus improving processing efficiency. Additionally, the model’s versatility extends beyond tracking, providing innovative solutions for other image processing tasks, including image recognition and object detection, showcasing its potential as a universal visual processing framework.

In order to thoroughly assess the performance of the proposed model, we constructed a multimodal aircraft tracking dataset (MATD) that encompasses two modalities that simulate the challenges of tracking aircraft in various environments, including low-light scenarios and complex backgrounds. This meticulously annotated dataset provides a comprehensive benchmark for evaluating the model’s tracking performance and offers invaluable resources for future research endeavors. Experimental results have demonstrated the superior tracking capabilities of MATI under extreme conditions, showcasing its adaptability to fluctuating environmental conditions and sensor limitations, thereby making a significant contribution to the field of visual tracking and enhancing aviation safety and surveillance technologies.

Extensive experiments conducted on our proprietary dataset, as well as other public datasets, validated the model’s effectiveness across a multitude of challenging scenarios, marking a substantial advancement in the domain of object-tracking technology. This research holds significant implications not only for researchers and engineers in the field of object tracking but also for the broader scientific community, providing an opportunity to deeply understand the potential and challenges of cross-spectral image fusion technology in practical applications. We anticipate that this study will provide robust technical support and develop new research directions for addressing key issues in multisource image fusion and adaptive object tracking.

The manuscript begins with a review of the literature in Section 2, distinguishing between single-modal and multimodal tracking technologies. Section 3 introduces our proposed network architecture, detailing innovative mechanisms and modules, and elaborates on the theoretical model and evaluation metrics we employ. This section also describes the Multimodal Aircraft Tracking Dataset utilized for our experiments. Section 4 presents the experimental setup and results, including a comparative analysis and ablation studies that highlight the effectiveness of specific model components. The discussion in Section 5 synthesizes these findings, discussing the key advancements and critically assessing the modules’ contributions to tracking performance. We conclude with considerations for future research that could further advance the field.

## 2. Related Works

### 2.1. Single-Modal Tracking

Correlation filter-based algorithms have gained traction in video object tracking due to their efficiency in handling accuracy and speed. Key advancements include addressing boundary effects with spatial regularization and background-aware strategies developed by Danelljan et al. [2] and Galoogahi et al. [3]. Additionally, the adaptation of Siamese networks, exemplified by SiameseFC by Bertinetto et al. [4] and SiameseRPN by Li et al. [5], incorporates region proposal networks that enhance tracking capabilities, especially for deformable targets. Building further on correlation filter enhancements, the State-aware Anti-drift Robust Correlation Tracking (SAT) by Han et al. [6] introduces a method that integrates environmental context and a color-based reliability mask to refine correlation responses under internal disturbances.

Deep learning models have also been influential, with MDNet by Nam et al. [7] utilizing domain-specific layers for target representation, though at reduced speeds. Enhanced robustness and adaptation to appearance changes are further achieved through multi-task frameworks such as ATOM by Danelljan et al. [8] and memory networks such as MemTrack by Yang et al. [9].

The transformer architecture, which was initially used for machine translation, has been widely adopted in the field of natural language processing (NLP) due to its efficiency. This architecture relies on attention mechanisms consisting of stacked encoder and decoder layers, each containing self-attention and a feed-forward network. The transformer by Vaswani et al. [32] converts input sequences into vectors and adds positional information through positional embeddings. Following the success of transformers in NLP tasks, several studies have explored their application to computer vision tasks. The TransT model by Chen et al. [10] adaptively combines the features of a template and a search area through the self-attention mechanism of the transformer. Wang et al. [11] connected templates and search areas across multiple frames using a transformer architecture and modeled the target’s invariant features for robustness by using the network’s self-attention mechanism. Yan et al. [12] utilized a transformer to model the spatiotemporal correlations between a template and search area, predicting the object’s location without the need for any anchors. Lin et al. [13] introduced a novel motion token in the transformer decoder, embedding motion information to enhance tracking robustness. Among these studies, the ViT model by Dosovitskiy et al. [33] has stood out. When trained on large datasets, ViT demonstrates superior results compared to state-of-the-art convolutional neural networks. Additionally, the authors of ViT tested it on medium and small datasets through fine-tuning, showcasing promising results. Following its success in image recognition, several ViT models have been developed and applied to other computer vision tasks, such as video object tracking. Ye et al. [14] developed OSTrack, which combines the feature learning and feature fusion processes from the ViT backbone. They discovered that some tokens from the search image contain background information, which is unnecessary to include during the tracking process. Based on this fact, OSTrack incorporates an early candidate elimination module in some encoder layers to remove tokens containing background information. By effectively utilizing the information flow between the target template and search area features, extracting target-specific discriminative cues, and eliminating unnecessary background features, OSTrack has demonstrated excellent tracking performance and speed in benchmark datasets.

### 2.2. Multimodal Tracking

Multimodal tracking technologies enhance adaptability and processing capabilities in complex environments by integrating data from different sensors. Despite challenges such as limited dataset sizes, researchers have advanced multimodal tracking through pre-training and fine-tuning strategies, as well as the introduction of efficient prompt learning methods. In the field of RGB-T (red, green, blue, and thermal infrared fusion) tracking, significant progress has been made in recent years.

In order to deploy the complementarity of the features at all hierarchical levels, Zhu et al. [15] proposed a recursive strategy for the dense aggregation of these features, resulting in robust representations of the target object in each modality. The mfDiMP by Zhang et al. [16] embeds the multimodal feature connection process into the powerful DiMP tracking framework for RGB-T tracking. Zhang et al. [17] proposed a late fusion method to obtain global and local weights in multimodal fusion, considering both appearance and motion information and dynamically switching between appearance and motion cues. The SiamCDA by Zhang et al. [18] introduced a complementarity-aware multimodal feature fusion module that first reduces the modality differences between single-modal features and then fuses them to enhance the discriminability of the fused features. The SiamCSR network by Guo et al. [19] is an efficient multimodal tracking algorithm designed specifically for processing RGB and thermal infrared (TIR) data to improve tracking robustness under extreme illumination and thermal crossover conditions. This algorithm employs a dual-stream Siamese network architecture, introducing a dual-modality region proposal network and selection strategy to enhance the accuracy of the predicted bounding box. FANet by Zhu et al. [20] fuses deep features at different levels under a single modality, suppressing unimodal noise based on modality reliability and fusing features from different modalities; additionally, the MANet by Li et al. [21] expands the network’s width by adding extra modality-specific branches and fuses modality features in a simple yet effective manner. Li et al. [22] proposed an algorithm that enhances the robustness of weight calculation by ranking the modalities. Lan et al. [23] introduced a discriminative learning framework specifically designed to suppress background interference from different modalities to obtain modality-invariant target discriminative features. Late fusion is performed after obtaining the results from different branches, followed by various fusion methods to produce the final outcome. Zheng et al. [24] proposed a method with which to judge feature quality through scoring for late fusion at the scoring level. Zhang et al. [25] proposed an adaptive decision fusion (ADF) module and utilized a self-attention mechanism to fuse multimodal response maps. The BAT by Cao et al. [26] achieves crossmodal information complementarity through a universal bidirectional adapter, effectively integrating information from different sensors. This method implements bidirectional feature prompting between transformer layers through a lightweight hourglass structure, providing stable tracking performance under varying environmental conditions. In addition, Luo et al. [27] proposed an RGB-T tracking framework based on bidirectional prompt learning. This framework achieves the complementary fusion of different modality images at the feature extraction stage through a lightweight prompter working in two dimensions, transmitting information between modalities at a low computational cost. Recently, Hui et al. [28] introduced a module called TBSI, which enhances crossmodal interaction between RGB and TIR search areas by using the template as a bridge to aggregate and distribute target-related object and contextual information. This approach not only improves tracking accuracy but also updates the original template to include rich multimodal context collected from the template medium. Expanding upon this, Sun et al. [29] introduced a transformer-based RGBT tracking system employing spatio-temporal multimodal tokens, which integrates dynamic tokens from previous frames to enhance tracking accuracy and responsiveness in real-time applications. Furthermore, Ding et al. [30] have developed a framework that utilizes an ’X Modality’ to enrich the feature space for RGBT tracking, aiming to improve the system’s robustness under various environmental challenges. Additionally, Zhang and Demiris [31] have pioneered a self-supervised RGB-T tracking method that leverages cross-input consistency to train on unlabeled video pairs, significantly reducing the dependency on manually annotated data while still achieving competitive tracking performance.

## 3. Materials and Methods

### 3.1. Network Architecture

In this section, we introduce our adaptive mechanisms designed for multimodal tracking, which are central to enhancing tracking accuracy and robustness across varied sensor modalities. The architectures of the crossmodal compensation (CMC) module and the token spatial filtering (TSF) module are then detailed. These components collectively form the backbone of our proposed method, the efficacy of which is illustrated in the system framework shown in Figure 1.

#### 3.1.1. Adaptive Mechanisms for Multimodal Tracking

Our proposed adaptive multimodal image object-tracking model significantly enhances the accuracy and robustness of visual target tracking, incorporating a vision transformer as its backbone network. While the model is capable of handling images from various sensors, including solely infrared images, our experimental focus was not on tracking with only infrared imagery. As such, our experimental section primarily presents results for visible light alone and in combination with infrared images. The model operates efficiently under varying environmental conditions, demonstrating its exceptional flexibility. It seamlessly processes a variety of input image sizes and types, primarily focusing on registered sets of visible light and combined visible-infrared images, as these configurations were empirically tested and emphasized in our research.

The model operates efficiently through an internal adaptive adjustment mechanism. Position embedding dynamically adjusts according to the size and structure of the input images, ensuring the accurate localization of each image patch within the model. Additionally, an adaptive attention mechanism tailored to the features of the input images enhances focus on target areas and effectively minimizes background distractions.

In practical scenarios, modal data incompleteness is a common issue due to limitations of imaging equipment or environmental factors. For example, infrared images may be unobtainable in extreme weather conditions, or the quality of visible light images may significantly degrade under certain lighting conditions. Existing multimodal visual tracking methods [25,26,27,28,29,30,31] typically assume that all modal data are available, an assumption often unrealistic in real-world applications. This fundamental presumption restricts the applicability of these techniques in complex or unstable environments. Our model innovatively addresses this challenge by maintaining efficient tracking performance even when some modal data are missing. Through a highly adaptive processing mechanism, our model dynamically adjusts its strategies based on the types and quality of available data, significantly enhancing the accuracy of target tracking and the overall robustness of the system.

In dual-modal tracking tasks, the model guides data flow through specially designed CMC modules that process and co-ordinate template features from different modalities, enhancing target recognition and localization. This proactive optimization strategy significantly improves tracking accuracy and robustness in dual-modal environments. In single-modal tracking tasks, the model intelligently bypasses the CMC modules, processing single-modal data directly to avoid unnecessary computational overhead. This autonomous decision-making process demonstrates the model’s control over its network structure, ensuring efficient tracking performance, even with limited resources.

This innovative design not only excels in multimodal tracking tasks but also maintains high processing efficiency in single-modal scenarios. The adaptive network structure remains stable under changing environmental conditions and sensor limitations, greatly enhancing the model’s practicality and applicational range. Additionally, the model’s flexibility extends to other image processing tasks, such as image recognition and object detection, further demonstrating its potential as a versatile visual processing framework.

#### 3.1.2. Crossmodal Compensation Module

In the domain of multimodal tracking, the design of the Crossmodal Compensation (CMC) module is founded on the principle of integrating feature extraction with relational modeling to facilitate a seamless flow of information between distinct modalities. This module is pivotal in handling tasks where both visible light and infrared images are processed, performing sophisticated transformations that ensure coherence and complementarity in the representations of multimodal templates. Unlike existing methodologies such as the X Modality [30], which employs the Pixel-Level Generation Module (PGM), Feature-Level Interaction Module (FIM), and Decision-Level Refinement Module (DRM) primarily for feature generation and refinement, our CMC module utilizes a dynamic crossmodal attention mechanism. This not only adjusts but significantly enriches template features by integrating the relational context between visible and infrared spectrums, thereby enhancing the adaptability and robustness of the tracking system.

Diverging from the BAT [26], which emphasizes dimensionality manipulation by separately encoding and decoding features across modalities before combining them, our CMC module adopts a more interconnected approach through crossmodal attention. This mechanism ensures that features from different modalities are not just combined but are dynamically influenced by each other’s contextual properties. Such an approach facilitates deeper integration and real-time adaptation to changes, providing superior tracking performance in complex scenarios. Moreover, unlike the TBSI [28], which statically utilizes template information to guide the search process, our model employs a dynamic attention-based mechanism that adapts to evolving scene contexts. This allows for more flexible and effective integration of modalities across both spatial and temporal dimensions, setting a new standard in the field of advanced tracking systems.

For the task at hand, the module accepts inputs from the visible spectrum features $x_{v}\in R^{C\times H\times W}$ and the infrared spectrum features $x_{i}\in R^{C\times H\times W}$, where *C*, *H*, and *W* denote the channels, height, and width of the feature maps, respectively. The infrared features are projected into a query space Q using a transformation matrix $E_{Q}\in R^{C\times D}$. This matrix characterizes the transformation from the original feature dimensions C×H×W to a reduced dimensionality C×D, optimizing the feature representation for subsequent processing steps:(1)$$Q=x_{i}E_{Q},\;E_{Q}\in R^{C\times D}$$

Concurrently, the visible light features are mapped into the key, K, and value, V, spaces using a similar linear transformation defined by $E_{KV}$:(2)$$K=V=x_{v}E_{KV},\;E_{KV}\in R^{C\times D}$$

Thereupon, a multi-head attention mechanism elucidates the interaction between modalities through the computation of attention outputs, O, and weights, A, fundamentally described as the following:(3)$$A=\mathrm{Softmax}\left(\frac{QK^{T}}{\sqrt{d_{k}}}\right),\;O=A\;\cdot \;V$$

The attentive outputs capture the crossmodal interactions, where the learned attention weights, A, emphasize the salient features in the infrared modality that are most relevant to the visible modality features.

The attention outputs, O, are subsequently refined through a feed-forward network (FFN), consisting of layers with nonlinear activation functions. This feed-forward network introduces additional complexity and abstraction to the feature representation, allowing for a richer and more nuanced integration of the modalities.

Ultimately, a layer normalization step combined with residual connection encapsulates the feature enhancement process:(4)$$H=\mathrm{LayerNorm}(x_{v}+O^{\prime })$$

This final step ensures that the augmented feature representation, H, is both normalized and stabilized, encapsulating the adaptive fusion of the visible and infrared modalities. The output, H, therefore, symbolizes a refined synthesis of features poised for subsequent tracking tasks, now enriched with crossmodal contextual insights.

By fostering such intermodal compensation, the CMC module’s architecture serves as a testament to the viability of leveraging attention mechanisms for concurrent feature extraction and relation modeling within the realm of multimodal object tracking.

#### 3.1.3. Token Spatial Filtering Module

By building on the principles established by prior advancements [14], the token spatial filtering (TSF) module introduces a refined focus on the spatial orientation of attention weights within the attention mechanism. This focus not only differentiates our work from preceding efforts but also infuses the tracking model with an intrinsic spatial awareness that more precisely reflects the target’s locational nuances.

Central to the operation of the TSF module is the methodical evaluation of the attention map, A, which discerns and prioritizes tokens based on their spatial relevance to the target area. This evaluation is achieved through an attentive curation process. The attention map for the target and search regions is defined as the following:(5)$$A_{t,s}=A\left[:,:,:L_{t},L_{t}:\right],\;T=\left[T_{t};T_{s}\right]$$

Here, $A_{t,s}$ represents the attention weights specific to the target and search spaces and T is the combined tensor of tokens from both the template, $T_{t}$, and the search, $T_{s}$, regions. $L_{t}$ represents the length of tokens from the template regions.

Upon obtaining the mean attention across the template and search regions, a sorting operation yields indices that highlight the spatially pivotal areas. This step is described by the following equation:(6)$$A_{\mathrm{mean}}^{\prime }=\mathrm{mean}\left(A_{t,s},\dim=2\right),\;\mathrm{indices}=\mathrm{argsort}\left(A_{\mathrm{mean}}^{\prime },\mathrm{descending}\right)$$

In this equation, $A_{\mathrm{mean}}^{\prime }$ denotes the mean attention across spatial locations, and indices are the sorted indices corresponding to the most attentive regions.

From this sorted array, a subset of indices corresponding to the highest attention scores is extracted, encapsulated by the keep ratio ρ:(7)$$\rho =\mathrm{keep\_ratio},\;\mathrm{topk\_indices}=\mathrm{indices}[:,:\left\lceil \rho \times L_{s}\right\rceil ]$$

Here, ρ represents the proportion of indices to keep, which is determined by the keep_ratio, and topk_indices represents the selected indices of the top attention scores. $L_{s}$ represents the length of tokens from the search regions

In an innovative step that preserves spatial coherence post-filtering, the top-k indices undergo a re-ordering process. This re-ordering ensures that the spatial layout of the search feature tokens remains intact, maintaining the integrity of the target’s spatial information:(8)keep_indices=sort(topk_indices)

The spatially sorted tokens, represented as $T^{\prime }$, are recombined with the template tokens $T_{t}$ to form a spatially coherent and feature-enriched representation $T^{\prime \prime }$:(9)$$T^{\prime }=T⊙\mathrm{keep\_indices},\;T^{\prime \prime }=T_{t}⊕T^{\prime }$$

In the above equation, $T^{\prime }$ signifies the filtered set of tokens, while $T^{\prime \prime }$ indicates the final feature representation that combines the refined tokens with the template features. The ⊙ operator is used to select specific tokens from a set based on indices. The ⊕ operator is then used to concatenate two sets of tokens to form a unified feature set.

By incorporating this step of re-sorting the top-k indices, the TSF module not only ensures that the most informative tokens are selected but also that their spatial arrangement echoes the structural layout of the target within the scene. This level of spatial fidelity is what sets our model apart, offering enhanced tracking performance by dynamically concentrating on the critical areas of interest within the image space.

### 3.2. Theoretical Model and Evaluation Metrics for Simulating Device Behavior

When evaluating the performance of target tracking devices such as electro-optical theodolites, experiments typically rely on pre-tracked video footage, where targets are precisely positioned at the center of the frame to ensure tracking accuracy under ideal conditions. However, this experimental setup significantly differs from real-world applications, particularly in scenarios where the device needs to dynamically adjust its viewpoint based on the target’s movements. This study is dedicated to developing a theoretical model to simulate the tracking algorithm’s response to dynamic changes in target position, thereby assessing its adaptability and effectiveness in handling false detections and other tracking anomalies.

This model was specifically developed to analyze images collected by our electro-optical theodolite devices. The evaluation metrics derived from this model are used to assess the performance of our self-created long-sequence dataset, the multimodal aircraft tracking dataset (MATD), which encompasses both single-mode and dual-mode image sequences. This dataset will be discussed in further detail in Section 3.3.

#### 3.2.1. Device Characteristics

The key functionality of target tracking devices, such as electro-optical theodolites, is their ability to dynamically adjust the viewing angle based on the deviation of the target position from the center of the frame, known as the off-target amount. Devices typically continue tracking along the predicted trajectory of the target for a set number of frames to compensate for potential false detections. During false detections, even though the predicted co-ordinates of the target may deviate from the actual co-ordinates, the device can still temporarily maintain its current trajectory, providing an opportunity to re-acquire and accurately track the target again.

#### 3.2.2. Relationship Modeling and Construction

The core of this study lies in simulating scenarios of false detections in test videos to map the ideal target’s position in actual device usage. Specifically, by analyzing changes in the co-ordinates of falsely detected targets in test videos, we can deduce the potential shifts in the frame position of the ideal target in practical device applications. When a falsely detected target reaches the edge of the frame in a test video, it corresponds to a scenario where the ideal target would have moved out of or is about to move out of the device’s field of view. This analysis is based on the assumption that the positional changes of targets in test videos can simulate the device’s response in similar real-life scenarios.

Relative position calculation: Analyze the changes in distance of falsely detected targets from the center of the frame in the test videos. Use these data to infer the potential position of the ideal target relative to the frame center in actual device usage.

False detection and tracking recovery: By using real-time monitoring and recording the position of targets in each frame, determine any occurrences of false detections. Analyze the changes in the position of targets after a false detection and assess whether the target returns near the center of the frame within a set number of frames.

#### 3.2.3. Tracking Evaluation

In this section, we conduct a two-part evaluation of the tracking algorithm’s performance following a false detection event. Firstly, we assess whether the algorithm can reposition the target close to its true co-ordinates within a predetermined number of frames set by the device’s trajectory prediction. Secondly, we evaluate the ability of the tracking algorithm to maintain stability in the target’s position for a continuous set of frames after recovery from a false detection.

By using this methodology, we utilize pre-tracked test video data as a basis to simulate and rigorously evaluate the performance of our tracking algorithm under realistic dynamic scenarios and potential misidentification situations. Below, we provide detailed descriptions of the quantitative procedures we employed to assess the efficacy of the tracking algorithm:

#### Notation

$\left(x_{gt},y_{gt},w_{gt},h_{gt}\right)$: the ground truth bounding box co-ordinates, where $x_{gt}$ and $y_{gt}$ denote the top-left corner co-ordinates, and $w_{gt}$ and $h_{gt}$ represent the width and height, respectively.$\left(x_{pred},y_{pred},w_{pred},h_{pred}\right)$: the predicted bounding box co-ordinates, with $x_{pred}$ and $y_{pred}$ indicating the top-left corner.$\left(x_{center},y_{center}\right)$: the center co-ordinates of the bounding box, which are calculated as the midpoint between the bounding box corners.

#### Displacement Vectors

The displacement vectors for the ground truth ($d_{gt}\left(t\right)$) and predicted bounding boxes ($d_{pred}\left(t\right)$) are calculated to represent the displacement from the center of the bounding box:(10)$$\begin{array}{cc} d_{\mathrm{gt}}\left(t\right) & =\left(x_{gt}+\frac{w_{gt}}{2}-x_{center},y_{gt}+\frac{h_{gt}}{2}-y_{center}\right), \end{array}$$(11)$$\begin{array}{cc} d_{\mathrm{pred}}\left(t\right) & =\left(x_{pred}+\frac{w_{pred}}{2}-x_{center},y_{pred}+\frac{h_{pred}}{2}-y_{center}\right). \end{array}$$

The difference in the displacement vectors is given by the following:(12)$$\Delta d(t)=d_{\mathrm{pred}}\left(t\right)-d_{\mathrm{gt}}\left(t\right).$$

#### Threshold Parameters

$N_{i}:$ the maximum allowable displacement error threshold over a sequence of frames;$N_{s}:$ the accumulative error threshold over a time window;λ: the threshold that determines the acceptability of the displacement error.

#### Tracking Metrics

In order to quantitatively evaluate the performance of our tracking approach, we employ a tracking metric that incorporates both the displacement error and the bounding box overlap ratio. This composite metric is designed to offer a balanced assessment of tracking precision and reliability.

Single-frame evaluation (false detection assessment, $M_{t}$): This metric assesses whether the tracking algorithm erroneously identifies a frame as containing the target when it does not. It calculates the normalized displacement error against a predefined threshold, λ, to determine whether a false detection has occurred.
(13)$$M_{t}=\left\{\begin{array}{cc} 1 & \mathrm{if}\;\frac{\Delta d(t)}{\mathrm{Area}(B_{t})}>\lambda ,\\ 0 & \mathrm{otherwise}. \end{array}\right.$$

Here, $\mathrm{Area}(B_{t})$ represents the area of the ground truth bounding box.

Temporal robustness evaluation (recovery from false detection, $R_{t}$): This metric evaluates the tracking algorithm’s ability to correct its prediction within $N_{i}$ frames after a misidentification by assessing if the normalized displacement error reduces below the threshold, λ. A successful reduction indicates that the algorithm has effectively realigned with the true target position.
(14)$$R_{t}=\left\{\begin{array}{cc} 1 & \mathrm{if}\;\frac{\Delta d(t+N_{i})}{\mathrm{Area}(B_{t+N_{i}})}\leq \lambda ,\\ 0 & \mathrm{otherwise}. \end{array}\right.$$

Accumulative evaluation (stability post-recovery, $S_{t}$): This metric assesses the long-term stability and accuracy of the tracking algorithm by aggregating the displacement errors over $N_{s}$ frames. A score of 1 signifies that the algorithm has consistently maintained accurate tracking, demonstrating its reliability following any corrections.
(15)$$S_{t}=\left\{\begin{array}{cc} 1 & \mathrm{if}\;\sum_{i=t+1}^{t+N_{s}}\frac{\Delta d(i)}{\mathrm{Area}(B_{i})}\leq N_{s},\\ 0 & \mathrm{otherwise}. \end{array}\right.$$

Overall tracking success: the overall tracking success is determined by the following:(16)$$\mathrm{Tracking}\;\mathrm{Success}=\left(\sum_{t=0}^{n}M_{t}=0\right)\vee \left(\overset{n-N_{i}-N_{s}}{\underset{t=0}{⋀}}\left(M_{t}=1\to \left(R_{t}\wedge S_{t+N_{i}}\right)\right)\right).$$

Notes:The first condition implies successful tracking if no single frame exceeds the displacement error threshold;The second condition ensures that, if any frame meets the single-frame evaluation criterion, then subsequent frames must satisfy both robustness and accumulative evaluations.

### 3.3. Multimodal Aircraft Tracking Dataset

#### 3.3.1. Background

In practical engineering applications, tracking aircraft on carriers presents numerous challenges, especially under extreme conditions. For instance, during twilight hours and when an aircraft transitions into the landing phase, distinguishing it from the background becomes difficult, increasing the likelihood of target loss. Additionally, in high-altitude flights, the small size of the targets and potential interference from airborne objects in infrared imagery may lead to false detections. In order to address these specific challenges, we meticulously designed and collected a dataset that adequately reflects and simulates these real-world application scenarios.

#### 3.3.2. Dataset Description

The dataset used in this study was collected using electro-optical theodolites under various environmental conditions to capture images of carrier-based aircraft in diverse situations. The collection process encompassed variations in lighting over different times and a variety of complex backgrounds to ensure the representativeness and diversity of the dataset. The images in the dataset range from simple to complex environmental conditions and cover multiple motion modes and morphological changes of aircraft targets.

The dataset consists of 17 video sequences, including individual visible light images and matched visible-infrared image pairs, each sequence containing manually annotated tracking information. These video sequences encompass various stages of aircraft operations, such as take-off, landing, and taxiing, totaling tens of thousands of frames, with 19,580 frames undergoing rigorous preprocessing. All collected image sequences were subjected to quality enhancement and noise reduction preprocessing steps to improve data usability. Each target’s location within the sequences was precisely annotated to ensure the accuracy of the tracking tasks.

#### 3.3.3. Dataset Configuration

In this study, our goal was to construct a custom image dataset specifically for training deep learning models on aircraft target tracking by extracting frames from lengthy video sequences. These video sequences cover the complete process from ground movement and take-off ascent to high-altitude flight, providing a rich variety of scenes and target behaviors. In order to ensure broad coverage and diversity in target size, posture, and phase, we employed a stratified sampling method. Specifically, the dataset was divided into training, validation, and test sets in proportions of 70%, 15%, and 15%, respectively. This division strategy is designed to ensure that the model learns features from different stages during training while maintaining good generalization capabilities during the validation and testing phases. In order to enhance the model’s adaptiveness and robustness to complex scenes, we implemented a series of image preprocessing and enhancement measures:low-light processing: Given that the dataset already includes images from different lighting conditions, to further enhance the model’s robustness under extreme low-light conditions, we intensively processed a subset of images. In each phase of flight—ground movement, take-off ascent, and high-altitude—25% of the image frames were selected randomly for low-light preprocessing. This strategy ensures an even distribution of low-light images throughout the dataset, helping to prevent model overfitting to specific lighting conditions and enhancing the model’s generalization to varying light environments. Low-light preprocessing techniques mainly include brightness adjustment to simulate images captured in inadequately lit environments and contrast enhancement to better distinguish targets from backgrounds amid low visibility.Data registration: In constructing the multimodal dataset, we emphasized the importance of imperfect data alignment. Our dataset is manually processed to ensure that targets in visible and infrared images are closely aligned, which is crucial for effective tracking. Although the shapes and sizes may not be perfectly consistent due to differences in camera optics and focal lengths, this type of imperfect registration is more representative of real-world multisensor imaging scenarios. This registration approach reflects our deep understanding of real-world multimodal data characteristics. In practice, differences in sensor installation, angles, and focal distances often result in variations in the images’ shape and size. By accurately aligning only the target positions, our model better adapts to this diversity, learning how to effectively track targets under varying sensor characteristics.Data annotation: Each frame of the visible and infrared sequences was meticulously annotated through two rounds of manual review, ensuring precise dataset annotations. We utilized the COCO [34] dataset’s annotation format, which includes the co-ordinates of each target’s bounding box (x, y, w, h). This standardization facilitates cross-dataset comparisons and aids in model transfer learning. Additionally, aligning with the GOT-10k [35] format, we incorporated several crucial annotation files to document target status and attributes. Absence.label indicates target visibility per frame, marking ’1’ for absence and ’0’ for presence. Cover.label quantifies the degree of occlusion on a scale from 0 (fully occluded) to 1 (fully visible), with an unobstructed view marked as 8. Cut_by_image.label identifies if a target is truncated by the image border.

## 4. Experiments and Results

### 4.1. Experiment Setup

#### 4.1.1. Experimental Environment and Datasets

Our experiments were conducted using the PyTorch framework, incorporating a mix of publicly available and custom-built datasets such as OTB-100 [36], GOT-10k [35], GTOT [37], RGBT234 [38], and our proprietary multimodal aircraft tracking dataset (MATD). For scaling and diversity considerations, 80% of the GTOT and RGBT234 data were used for training, with the remaining 20% allocated to performance testing. A detailed description of the MATD dataset is provided in Section 3.3. All experiments were executed on a single NVIDIA RTX 4090 GPU.

#### 4.1.2. Model Configuration and Training Details

We utilized a ViT-base [33] model, pretrained via the MAE [39] approach, and we employed the AdamW optimizer for parameter adjustments. The initial learning rate was set to 0.0004, with a weight decay coefficient of 0.0001, targeting stable parameter updates during training to effectively minimize overfitting. The batch size was established at 32 to balance training efficiency against memory consumption. After 80 training epochs, the learning rate was reduced following a stepwise strategy, with a decay rate of 0.1, aiming to further refine model performance in the later stages of training. The overall training was planned for 150 epochs based on preliminary experimental outcomes to ensure stable performance at the expected level.

#### 4.1.3. Input Data Handling

In our experimental setup, template sizes were fixed at 128 pixels, and search areas were set at 256 pixels. Given the variations in imaging sizes from our electro-optical theodolite equipment, the scaling factors for the visible light and infrared image templates were, respectively, set at 2.0 and 3.0. This configuration ensures that the model adapts to size variations between different modalities.

### 4.2. Comparative Study and Analysis of TSF Module

#### 4.2.1. Experimental Setup and Datasets

In order to evaluate the effectiveness of the token spatial filtering (TSF) module, we trained and tested both the baseline model [14] and the model integrated with the TSF module. The training dataset comprised primarily of GOT-10k and a subset of MATD. Testing was conducted on the GOT-10k, OTB-100, and MATD datasets.

#### 4.2.2. Performance Evaluation

As indicated in Table 1, although there was a slight performance drop on the GOT-10k test set with the introduction of the TSF module, significant improvements were observed on the non-training datasets, such as OTB-100 and MATD. The key performance metrics included the following:SR_0.5_, SR_0.75_, and SR_0.9_: these metrics represent the success rates at thresholds of 0.5, 0.75, and 0.9, respectively, indicating the proportion of tracking instances where the intersection over union (IoU) between the predicted and ground truth bounding boxes exceeds these thresholds.Average overlap ratio (AOR): this measures the average IoU between the tracking and ground truth bounding boxes.Average precision error (APE): this reflects the average error in the location of the predicted bounding boxes.

Specifically, for OTB-100, improvements of the following were recorded: 2% for SR_0.5_, 8% for SR_0.75_, and 15% for SR_0.9_, with a 2.6% increase in AOR and a 31% reduction in APE; for MATD, there was a 1% improvement in SR_0.5_, a 15% improvement in SR_0.75_, and an 18% improvement in SR_0.9_, with a 3.6% increase in AOR and a 30% reduction in APE.

#### 4.2.3. Visualization Analysis

The visualizations depicted in Figure 2 allow us to compare the attention distribution effects of integrating the TSF module into the model. In the baseline model visualization (Figure 2(a1–a3)), the attention is more dispersed, which suggests that the model is less effective at focusing specifically on the target and is more influenced by the surrounding background. Conversely, the TSF-integrated model (Figure 2(b1–b3)) shows a more concentrated attention pattern around the target, demonstrating the TSF module’s capability to enhance target focus and suppress background noise, thereby improving tracking accuracy.

##### Further Visualizations

Additional visualizations detailing the TSF module’s optimization of spatial filtering capabilities are available in Appendix A. This appendix provides further evidence of the module’s effectiveness in enhancing the accuracy and stability of target tracking in complex environments.

### 4.3. Ablation Study and Analysis of CMC Module

#### 4.3.1. Module Design and Integration in Vision Transformer (ViT) Backbone

In the development of our dual-mode image processing capabilities within the vision transformer (ViT) backbone, we designed two distinct modules: the more compact CMC module and the CMCe module, which includes a gating mechanism and has a larger parameter count. The inclusion of the CMCe module was intended to explore how gating mechanisms might regulate information flow and enhance feature representation. However, the experimental results did not consistently show an advantage over the simpler CMC module, allowing us to assess the trade-offs between enhanced complexity and computational efficiency.

Further integration strategies involved the strategic placement of these modules in the top layer of the ViT backbone. The CMC module in both its standard and enhanced (CMCe) forms was configured to follow the token spatial filtering (TSF) module. This setup was aimed at enhancing the co-ordination of multimodal data following spatial attention refinement. The preliminary tests of an alternative configuration, where the CMC module preceded the TSF module, yielded sub-optimal results in terms of tracking accuracy and stability. These findings led us to discontinue any further exploration of this configuration and instead focus on the more effective sequence, where the CMC module follows the TSF module. This arrangement ensures an optimized processing flow that effectively leverages the strengths of each module, demonstrating a practical approach to enhancing tracking performance within complex multimodal environments.

#### 4.3.2. Computational Efficiency Evaluation

A rigorous testing regime was conducted within a GPU environment equipped with NVIDIA CUDA to comprehensively evaluate the performance of our vision transformer-based deep learning models. Before the performance assessment, the model configurations and parameter settings were standardized to ensure experimental consistency and reproducibility. The template and search images were generated using a standard normal distribution. The PyTorch framework’s profiler tool calculated the model’s multiply accumulates (MACs) and parameter count (PARAMs) for specific inputs. In order to measure the model’s frames per second (FPS), an initial model warm-up was performed, processing a designated amount of data to stabilize GPU performance. Subsequently, the model was run continuously under controlled conditions, processing an adequate number of image frames and recording the total required time accurately.

The computational efficiency differences between CMC and CMCe configurations, as illustrated in Table 2, highlight the trade-offs involved in enhancing feature representation capabilities within these architectures. Notably, the integration of the CMCe module at the top of the network (indicated as “CMCe (top)” in Table 2) results in a modest increase in MACs by 2.6% and an increase in PARAMs by 8.9% compared to the baseline model configured for single-mode processing. Despite these increases, there is a corresponding decrease in FPS by approximately 6.2%, reflecting the additional computational burden introduced by the gating mechanisms of CMCe.

Further, when CMCe is combined with CMC in both the top and middle layers of the network (“CMCe+CMC (top and middle)”), there is a more pronounced increase in both MACs and PARAMs—7.9% and 28%, respectively. This configuration, while offering potentially richer feature interactions and enhanced tracking accuracy, comes at the cost of reduced FPS, highlighting a significant decrement in processing speed by 11%.

On the other hand, the standalone CMC configuration at the top layer (“CMC (top)”) showcases the most efficient use of resources among the enhanced configurations, with only a minimal impact on computational resources and a slight decrease in FPS. This configuration strikes an optimal balance between performance enhancement and computational efficiency, making it particularly suitable for applications where speed is critical but moderate improvements in tracking capabilities are desired.

#### 4.3.3. Performance Evaluation on Public Datasets

Performance tests on the CMC and CMCe on public datasets have been thoroughly documented, demonstrating a detailed comparison of different configurations applied to the RGBT234+GTOT datasets. These evaluations are crucial, as they highlight the practical implications of our theoretical enhancements and the adjustments made to real-world tracking performance. The initial model training was conducted on GOT-10k and single-mode image sequences from our MATD, spanning 150 epochs. By leveraging the modality-adaptive characteristics of our framework, further training was performed on the dual-mode image groups.

Initially, all modules were frozen; the subsequent training phases on the RGBT234 and GTOT datasets continued for an additional 50 epochs, utilizing various strategic configurations to optimize module performance. The results of these extensive tests are presented in Table 3 and Table 4, providing a comprehensive view of our findings.

##### Training Strategies

In order to adapt our model configurations to different tracking challenges, several training strategies were employed:Strategy A: only the top-layer CMCe and the intermediate TSF modules were unfrozen, with the CMC module added to the middle blocks.Strategy B: only the top-layer CMCe was unfrozen, with the CMC module added to the middle blocks.Strategy C: only the top-layer CMCe was unfrozen, with no additional module added.Strategy D: the top-layer CMC and intermediate TSF modules were unfrozen, with the CMC module added to the middle blocks.Strategy E: only the top-layer CMC was unfrozen, with the CMC module added to the middle blocks.Strategy F: only the top-layer CMC was unfrozen, with no additional module added.

##### Performance Metrics and Results

Table 3 showcases the outcomes of employing different configurations to the RGBT234+ GTOT dataset. The performance metrics used for evaluation were the following:Mean precision rate (MPR): this quantifies the accuracy with which a tracker can localize the target in terms of the predicted bounding box precision against the ground truth.Mean success rate (MSR): this measures the proportion of successful tracking instances whereby the predicted bounding box overlaps with the ground truth by a threshold commonly accepted in tracking benchmarks.

As documented in Table 3, the configurations employing both a top-layer CMCe and intermediate modules (Strategy A and D) tend to show varied effectiveness, with Strategy F (only top-layer CMC) achieving the highest MPR and MSR, suggesting that simpler configurations may sometimes yield better real-world performance. Table 4 presents a comparative analysis with other trackers, showcasing that our configurations, particularly “CMC only”— which scored the highest in both MPR and MSR—significantly outperform established competitors such as SiamCSR, MDNet, and ECO. This superior performance underscores the effectiveness of our model enhancements in challenging tracking scenarios.

#### 4.3.4. Performance Evaluation on Private Datasets

Our MATD was evaluated under extremely low-light conditions to showcase the enhancements achieved through the CMC and CMCe modules in single-mode tracking, as well as their comparative effectiveness in dual-mode configurations. By utilizing the theoretical model outlined in Section 3.2.3, we established empirical thresholds: $N_{i}=5$, $N_{s}=10$, and λ=0.02. The evaluations included the following:Tracking success (TS): this metric evaluates whether the tracking was successful across the entire image sequence. Refer to Equation (Equation 16) for detailed computation methods.Number of false detections (NFDs): counts the instances when the model erroneously identifies nontargets as targets during the tracking sequence. The criteria for determining a false detection are detailed in Equation (Equation 13).Relative co-ordinate offset proportion (RCOP): this metric evaluates the maximum value from a list of normalized displacement errors across all frames in a tracking sequence. Each frame’s displacement error, Δd(t), is normalized by the area of the target bounding box, $\mathrm{Area}\;(B_{t})$, as shown in Equation (Equation 13). RCOP is defined by the highest normalized ratio obtained during the sequence. Lower RCOP values are beneficial, indicating minimal deviation from the target’s ideal trajectory, thus reducing the need for manual adjustments and ensuring greater operational stability of the electro-optical theodolite systems.

Training was conducted on single-mode images for 150 epochs, followed by an additional 10 epochs each for training CMC alone and in combination with TSF under dual-mode conditions. Various configurations, including different placements of CMCe and CMC modules, were systematically explored to optimize tracking performance in adverse lighting conditions. In order to ensure a robust evaluation, each configuration was tested multiple times, and the results were averaged to comprehensively assess both performance consistency and stability.

As shown in Table 5, the optimal setups substantially decreased RCOP by 64%, addressing tracking challenges in low-light environments effectively. These results confirm the model’s robustness and the effectiveness of the dual-mode configuration, particularly with the inclusion of the CMC module, to enhance the tracking capabilities of our electro-optical theodolite devices.

In Figure 3, we visualized and compared the compensatory effects of the CMC module against configurations without it. The comparative visualization underscores the significant role of the CMC module in enhancing attention focusing within a neural network. The attention maps from layers 2, 7, and 9 clearly demonstrate that the integration of the CMC module results in a more concentrated and precise allocation of attention around the target area, particularly in the higher layers. This effect indicates that the CMC module effectively synchronizes and refines intermodal feature integration, thus substantially improving the model’s capability to accurately localize and track targets in complex visual environments.

Furthermore, tests were also conducted under normal lighting conditions without infrared support, where our model successfully tracked targets with an RCOP (%) of merely 0.34%, demonstrating robust performance across varying environmental settings.

## 5. Discussion and Conclusions

### 5.1. Overview of Key Findings

In this study, we integrated sophisticated neural network architectures and adaptive mechanisms such as the vision transformer (ViT), the crossmodal compensation (CMC) module, and the token spatial filtering (TSF) module to enhance multimodal image object tracking. These technologies significantly improved tracking precision and robustness across various environmental and operational scenarios. Due to our adaptive mechanisms for multimodal tracking, the system demonstrated strong performances in both single- and dual-mode scenarios. Additionally, under our theoretical model, the evaluation metrics indicated high effectiveness, confirming the success of our approach.

### 5.2. In-Depth Discussion of Modules

#### 5.2.1. TSF Module Performance Variations

The TSF module demonstrated slightly reduced performance on the training dataset’s test set but showed improved performance on unknown datasets. This divergence likely stems from the module’s design, which is tailored to enhance focus and suppress background noise effectively in diverse environments. While the training set may closely align with the conditions under which the model was optimized, unfamiliar datasets present new challenges and scenarios where the capabilities of the TSF module to generalize are more beneficial. This suggests that the TSF module enhances the model’s robustness to new conditions, a critical factor for real-world applications where environments can vary significantly.

#### 5.2.2. Challenges with the CMCe Module

Despite the CMCe having more parameters and introducing new gating mechanisms intended to optimize feature extraction and integration, its performance did not consistently surpass the simpler CMC module. This could be because the increased complexity of CMCe led to overfitting on training data, making it less effective on test datasets that differ from the training scenarios.

#### 5.2.3. Placement of CMC Modules

The configuration involving CMC modules placed in both the network’s top and middle layers showed mixed results: a better MPR but a worse MSR. This outcome can be attributed to the dimensional discrepancies in the feature inputs between the top and middle layers. While the top layer processes aligned features across modalities, the middle layers deal with features that may not be perfectly aligned, impacting the coherence of the feature integration. This misalignment can lead to less effective intermodal compensation, reducing the overall success rate despite potentially higher precision in specific scenarios.

### 5.3. Future Directions and Potential Enhancements

Looking ahead, we plan to refine the TSF module by enhancing its dynamic spatial focusing and improving robustness across more extreme conditions. These advancements will extend to other critical image processing tasks such as object detection and instance segmentation, where distinguishing between foreground and background is crucial. The TSF module, by reducing the number of runtime parameters, promises significant improvements in inference speed, providing a vital balance between speed and accuracy tailored to each specific task. Furthermore, the TSF module’s capability to dynamically adjust focus based on contextual information makes it highly suitable for instance segmentation tasks, helping distinguish between overlapping or closely situated objects by effectively managing the spatial resolutions at which these objects are processed.

Additionally, we aim to explore the implementation of CMC modules at every network layer, improving RGBT tracking performance through consistent feature alignment. The CMC module, by leveraging the complementary strengths of different modalities, can greatly enhance feature representation in object detection tasks where diverse environmental conditions may affect the visibility of objects. For instance, combining visible and infrared imagery can enable the detection system to perform reliably under various lighting conditions, enhancing detection robustness. Moreover, exploring the application of these modules in semantic segmentation can open new avenues for enhancing the granularity of segmentation. By dynamically adjusting to the scene’s complexity and integrating multimodal data, the TSF and CMC modules can help achieve more accurate and context-aware segmentation results, potentially reducing the computational overhead typically associated with processing high-resolution data. Exploring edge computing solutions to facilitate faster on-device processing, reduce latency, and increase the responsiveness of tracking systems in live environments is another promising direction. These enhancements and expansions will ensure our tracking systems not only meet the current demands of multimodal integration but are also well-prepared for future advancements in technology and application scenarios.

## Figures and Tables

**Figure 1 sensors-24-04911-f001:**
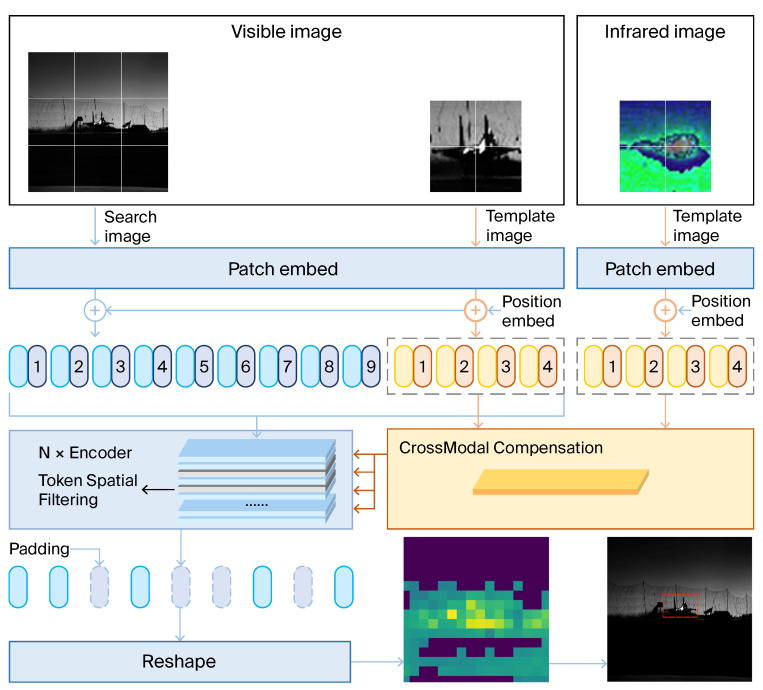
Schematic of the multimodal tracking network architecture. This diagram illustrates the comprehensive workflow of our tracking model, which processes both visible and optionally infrared image inputs. The visible and infrared images undergo initial patch embedding and subsequent position embedding. The data flow then progresses through a series of encoders, incorporating token spatial filtering. A key component, the crossmodal compensation module, is selectively engaged to integrate features from both image types, enhancing the system’s ability to handle diverse environmental inputs and improve target detection accuracy. This model architecture exemplifies our approach to adaptable, multimodal tracking by efficiently processing various input types and optimizing computational resources.

**Figure 2 sensors-24-04911-f002:**
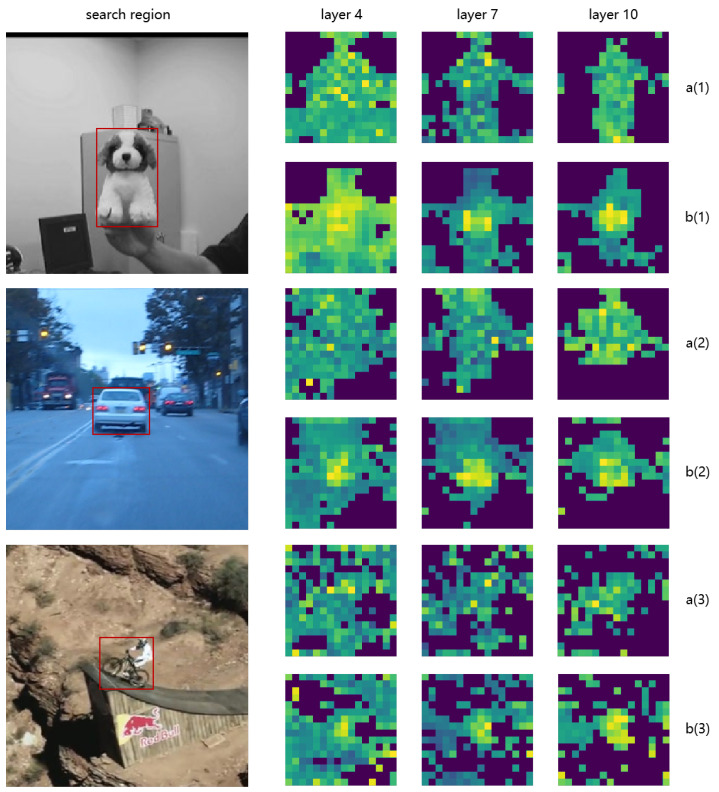
Visualizations of attention weights in the search areas corresponding to different tracking targets after various ViT layers. Subfigures labeled ’**a(1)**’, ’**b(1)**’, ’**a(2)**’, ’**b(2)**’, ’**a(3)**’, and ’**b(3)**’ represent attention distributions before and after employing TSF modules for targets 1, 2, and 3, respectively. Layers 4, 7, and 10 are displayed to illustrate how the model estimates similarity across positions in the search area. The red rectangles mark the locations of the target objects in each search region image.

**Figure 3 sensors-24-04911-f003:**
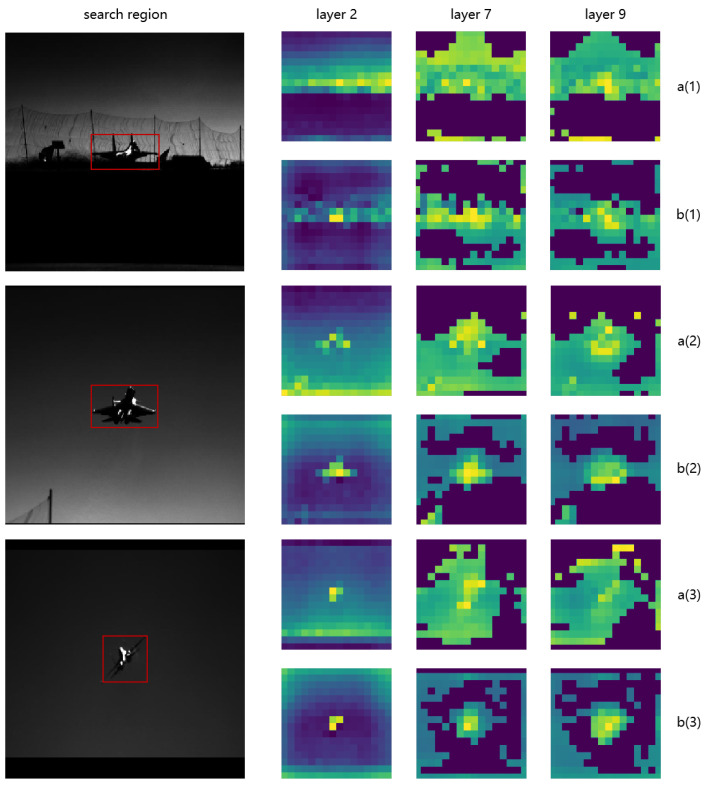
Visualization of attention maps at different layers across three flight phases, with and without the use of CMC modules. Subfigures labeled with ’**a(1)**’, ’**a(2)**’, ’**a(3)**’ represent the attention distribution before employing CMC modules during the first, second, and third flight phases, respectively. Correspondingly, ’**b(1)**’, ’**b(2)**’, ’**b(3)**’ depict the attention distribution after employing CMC modules. The red rectangle marks the location of the target object.

**Table 1 sensors-24-04911-t001:** Comparative analysis of performance metrics before and after the integration of the TSF module. Directional arrows: ↑ signifies improvements, and ↓ signifies reductions in performance metrics. The terms pre/post denote models without/with TSF integration, respectively.

Dataset	SR_0.5_	SR_0.75_	SR_0.9_	AOR	APE
	Pre/Post	Pre/Post	Pre/Post	Pre/Post	Pre/Post
GOT-10k	0.95/0.95	0.86/0.85 (−1%↓)	0.58/0.53 (−8.6%↓)	0.85/0.84 (−1%↓)	26.09/26.90 (+3%↓)
OTB-100	0.93/0.95 (+2%↑)	0.63/0.68 (+8%↑)	0.13/0.15 (+15%↑)	0.76/0.78 (+2.6%↑)	8.57/5.88 (−31%↑)
MATD	0.98/0.99 (+1%↑)	0.74/0.85 (+15%↑)	0.38/0.45 (+18%↑)	0.83/0.86 (+3.6%↑)	9.57/6.71 (−30%↑)

**Table 2 sensors-24-04911-t002:** Computational efficiency of CMC vs. CMCe.

Configuration	MACs (G)	Change	PARAMs (M)	Change	FPS
Base (single-mode)	21.52	-	92.12	-	221.97
CMCe+CMC (top and middle)	23.22	+7.9%	118.11	+28%	197.74
CMC+CMC (top and middle)	23.07	+7.2%	115.74	+26%	200.57
CMCe (top)	22.08	+2.6%	100.39	+8.9%	208.22
CMC (top)	21.93	+1.9%	98.03	+6.4%	216.95

**Table 3 sensors-24-04911-t003:** Performance of CMC and CMCe on RGBT234+GTOT. The best results are highlighted in bold. Thresholds for MPR and MSR were set at 20 and 0.5, respectively.

Configuration	MPR	MSR
CMCe+TSF+CMC	0.69	0.57
CMC+TSF+CMC	0.73	0.63
CMCe+CMC	0.72	0.64
CMC+CMC	0.75	0.68
CMCe only	0.76	0.68
CMC only	**0.78**	**0.71**

**Table 4 sensors-24-04911-t004:** Comparative tracker analysis on RGBT234+GTOT. The best results are highlighted in bold. Thresholds for MPR and MSR were set at 20 and 0.5, respectively.

	SiamCSR [19]	MDNet [7]	C-COT [40]	ECO [41]	SOWP [42]	SRDCF [2]	CSR-DCF [43]	KCF [44]	Ours
MPR	0.76	0.72	0.71	0.70	0.70	0.64	0.62	0.46	**0.78**
MSR	0.67	0.62	0.62	0.61	0.60	0.56	0.55	0.40	**0.71**

**Table 5 sensors-24-04911-t005:** Testing on MATD with CMC and CMCe modules. The best results are highlighted in bold. Thresholds for MPR and MSR were set at 10 and 0.75, respectively.

Configuration	TS	NFDs	RCOP (%)	MPR	MSR
Single-mode	False	-	3.12	0.69	0.44
CMCe+CMC	True	0	1.7	0.77	0.62
CMC+CMC	True	0	**1.12**	**0.92**	0.85
CMCe only	True	0	1.51	0.75	0.57
CMC only	True	1	1.41	0.91	**0.87**

## Data Availability

The study includes a dataset named MATD, which contains military-sensitive information and cannot be publicly shared due to national security. Despite this limitation, we have ensured that the research findings are reproducible and validated through stringent analysis. For inquiries on the methodology or general findings, the corresponding author can be contacted for information that adheres to confidentiality and security protocols.

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
