# Peer review of "MATI: Multimodal Adaptive Tracking Integrator for Robust Visual Object Tracking"

_sensors, 2024, doi:10.3390/s24154911_

Round 1

Reviewer 1 Report

Comments and Suggestions for Authors

The manuscript titled 'MATI: Multimodal Adaptive Tracking Integrator for Robust Visual Object Tracking' introduces an innovative adaptive multimodal image object-tracking model that significantly advances the state-of-the-art in visual tracking technology. The authors have developed a model that leverages the strengths of multi-spectral imagery, integrating infrared and visible light to enhance tracking accuracy and robustness under challenging conditions. The paper is well-structured, presenting a clear problem statement, a thorough literature review, and a detailed explanation of the proposed model's architecture and mechanisms. The comprehensive experiments conducted across various datasets, including a meticulously created multimodal aircraft tracking dataset (MATD), demonstrate the model's superior performance in adapting to rapid environmental changes and sensor limitations. The results showcase the potential of the MATI model to provide robust tracking solutions for applications such as earth observation and environmental monitoring, where tracking dynamic objects like aircraft is crucial amidst complex backgrounds and low-light conditions.

Comment 1:

   The paper introduces an adaptive multimodal image object-tracking model that integrates multi-spectral image sensors to enhance tracking accuracy and robustness by combining infrared and visible light imagery. The authors are advised to further clarify the innovative aspects of their model compared to existing technologies, particularly regarding the technical contributions in multimodal data processing and adaptive tracking strategies.

Comment 2:

The paper presents a theoretical model and evaluation metrics to quantify the performance of the tracking algorithm. The authors are encouraged to discuss in detail the rationale behind the selection of these metrics and how they comprehensively reflect the algorithm's performance in practical applications, as well as their comparison with existing evaluation methods.

Comment 3:

The discussion section of the paper should provide an in-depth analysis of the experimental results, comparison with existing studies, and identification of the study's limitations and future directions. It is recommended to expand the discussion, especially regarding the performance variation of the CMC and TSF modules under different conditions and how these modules can be applied to other image processing tasks.

Comment 4:

The paper reviews several studies related to visual object tracking. It is suggested to ensure the comprehensiveness of the literature review by considering the latest research in recent years, showcasing the most recent developments in the field. At the same time, it is recommended to update the references to ensure that the most current research findings in the field are cited, enhancing the timeliness and academic contribution of the paper.

Comments on the Quality of English Language

The language needs serious revision

Author Response

Comment 1:

   The paper introduces an adaptive multimodal image object-tracking model that integrates multi-spectral image sensors to enhance tracking accuracy and robustness by combining infrared and visible light imagery. The authors are advised to further clarify the innovative aspects of their model compared to existing technologies, particularly regarding the technical contributions in multimodal data processing and adaptive tracking strategies.

Response 1:

Thank you very much for your insightful comments. Following your suggestions, I have enhanced the introduction and methods sections of the paper to include a more detailed comparison with existing technologies. Specifically, these comparisons have been updated and elaborated upon on page 1, lines 31 to 43; page 5, lines 207 to 218; and page 6, lines 233 to 256. I appreciate your guidance in making these improvements, which I believe now more clearly articulate the innovative aspects and technical contributions of our model.

Comment 2:

The paper presents a theoretical model and evaluation metrics to quantify the performance of the tracking algorithm. The authors are encouraged to discuss in detail the rationale behind the selection of these metrics and how they comprehensively reflect the algorithm's performance in practical applications, as well as their comparison with existing evaluation methods.

Response 2:

Thank you very much for your valuable feedback. The theoretical model was developed specifically to simulate the behavior of our electro-optical theodolite equipment in actual tracking scenarios, and the evaluation metrics were designed to assess the performance of the algorithm on this type of equipment. The key functionality of devices like electro-optical theodolites is their ability to dynamically adjust the viewing angle based on the offset amount, ensuring that the target remains centered in the frame. The device typically continues to track along the predicted trajectory of the target for a certain number of frames to accommodate potential misdetections. This approach to evaluation is tailored specifically for such tracking devices and is not as universally applicable as other evaluation methods. Regarding existing evaluation methods, we have also conducted tests on public datasets, as detailed in section 3.2, starting at line 322 on page 8, as well as in sections 4.2.2 and 4.3.3, on lines 514 and 596 of pages 13 and 16, respectively. I appreciate your suggestions and believe this explanation may clarify the selection and application of our evaluation metrics.

Comment 3:

The discussion section of the paper should provide an in-depth analysis of the experimental results, comparison with existing studies, and identification of the study's limitations and future directions. It is recommended to expand the discussion, especially regarding the performance variation of the CMC and TSF modules under different conditions and how these modules can be applied to other image processing tasks.

Response 3:

Thank you for your constructive feedback. I have addressed the performance variations of the CMC and TSF modules under different conditions in section 4.2 on page 12, line 508; section 4.3 on page 14, line 544; and section 5.2 on page 19, line 691. Additionally, the limitations of the study and future directions have been discussed in the discussion section, where I have also expanded on how these modules can be applied to other image processing tasks. Please refer to lines 716 to 743 on page 20 for a detailed discussion. I appreciate your suggestions and believe these additions will enrich the paper by providing a deeper insight into the capabilities and potential applications of our modules.

Comment 4:

The paper reviews several studies related to visual object tracking. It is suggested to ensure the comprehensiveness of the literature review by considering the latest research in recent years, showcasing the most recent developments in the field. At the same time, it is recommended to update the references to ensure that the most current research findings in the field are cited, enhancing the timeliness and academic contribution of the paper.

Response 4:

Thank you for your valuable suggestions. I have updated the references section to include the latest research, which reflects the most recent developments in the field of visual object tracking. Specifically, please refer to lines 174 to 182 on page 4 for the updated references, which now include:

[1] Sun D, Pan Y, Lu A, et al. Transformer RGBT Tracking with Spatio-Temporal Multimodal Tokens. arXiv preprint arXiv:2401.01674, 2024.

[2] Ding Z, Li H, Hou R, et al. X Modality Assisting RGBT Object Tracking. arXiv preprint arXiv:2312.17273, 2023.

[3] Zhang X, Demiris Y. Self-Supervised RGB-T Tracking with Cross-Input Consistency. arXiv preprint arXiv:2301.11274, 2023.

I appreciate your guidance, which has helped enhance the academic rigor and relevance of my paper.

Reviewer 2 Report

Comments and Suggestions for Authors

This paper proposes a multimodal object tracking algorithm. However, there are still some issues that need to be considered.

1. The introduction should be expanded to include more literature on solutions by other researchers for handling environmental disturbances such as fast motion, appearance variations, atmospheric interference, biological activity, and ground clutter, to demonstrate the research progress made by other scholars in addressing these challenges.

2. In line 180 of the paper, it is mentioned that the proposed model can seamlessly processing various input image sizes and types, including individual visible light images, infrared images, and their combinations. However, the description in Figure 1 indicates that the model can process visible light and optional infrared image inputs. The experimental section only presents results for visible light and visible light + infrared, without any single infrared modality experiments.

3. In equations (1) and (2), the explanation for R^(C*D) is missing.

4. In equations (5) and (6), the explanations for L_t and L_s  are missing.

5. In equation (9), the explanation for the computation symbol  and âŠ™ is missing.

6. In Figures 1 and 3, the images used appear to be of fighter jets. Please confirm if the use of these images is appropriate for the content of the paper.

7. Some of the relevant literature should be reviewed and discussed, namely [1].

[1] "State-Aware Anti-Drift Object Tracking," in IEEE Transactions on Image Processing, vol. 28, no. 8, pp. 4075-4086, Aug. 2019, doi: 10.1109/TIP.2019.2905984.

Comments on the Quality of English Language

The English Editing seems to be okay.

Author Response

Comment 1. The introduction should be expanded to include more literature on solutions by other researchers for handling environmental disturbances such as fast motion, appearance variations, atmospheric interference, biological activity, and ground clutter, to demonstrate the research progress made by other scholars in addressing these challenges.

Response 1:

Thank you for your insightful comment. I have expanded the introduction to include a broader range of literature addressing environmental disturbances such as fast motion, appearance variations, atmospheric interference, biological activity, and ground clutter. These additions are detailed from line 31 on page 1 to line 43 on page 2. Furthermore, I have thoroughly discussed the contributions of other researchers in the related works section, with additional references now included from line 174 to 182 on page 4. I appreciate your suggestions, which have significantly enhanced the context and depth of the literature review in my paper.

Comment 2. In line 180 of the paper, it is mentioned that the proposed model can seamlessly processing various input image sizes and types, including individual visible light images, infrared images, and their combinations. However, the description in Figure 1 indicates that the model can process visible light and optional infrared image inputs. The experimental section only presents results for visible light and visible light + infrared, without any single infrared modality experiments.

Response 2:

Thank you for pointing out the ambiguity in my description. I intended to convey that infrared images can also be used as a single modality input, and our equipment and algorithms are capable of tracking targets in infrared alone. However, our research does not focus on tracking with only infrared imagery, which is why there were no experiments or comparative studies involving solely infrared images. To avoid any misunderstanding, I have revised the description accordingly. Please refer to lines 192 to 201 on page 5 for the updated text. I appreciate your feedback, which helps clarify this aspect of our study.

Comment 3. In equations (1) and (2), the explanation for R^(C*D) is missing.

Response 3:

Thank you for highlighting this oversight. I have now added an explanation for R^(C*D) in the manuscript. For detailed information, please refer to lines 259 to 263, spanning from page 6 to page 7. I appreciate your guidance, which has helped improve the clarity and completeness of the mathematical formulation in the paper.

Revisions: This matrix characterizes the transformation from the original feature dimensions C×H×W to a reduced dimensionality C×D

Comment 4. In equations (5) and (6), the explanations for L_t and L_s  are missing.

Response 4:

Thank you for pointing out this omission. I have clarified in the manuscript that L_t and L_s represent the lengths of the token sequences for the template and search features, respectively. For a detailed explanation, please refer to line 297 on page 7 and line 307 on page 8. I appreciate your attention to detail, which helps ensure the clarity and accuracy of our mathematical descriptions.

Comment 5. In equation (9), the explanation for the computation symbol ⊕ and ⊙ is missing.

Response 5:

Thank you for your observation regarding the missing explanations for the computation symbols

⊕and⊙in equation (9). I have now added a clear explanation for these symbols in the manuscript. 

Revisions: The⊙operator is used to select specific tokens from a set based on indices. The⊕operator is then used to concatenate two sets of tokens to form a unified feature set.

Comment 6. In Figures 1 and 3, the images used appear to be of fighter jets. Please confirm if the use of these images is appropriate for the content of the paper.

Response 6:

Thank you for your attention to the images used in Figures 1 and 3. I can confirm that these images are indeed of carrier-based fighter jets, which are part of our experimental dataset. Their use in the paper is solely for illustrative purposes to demonstrate the application of our model in tracking high-speed objects, and is therefore appropriate. I appreciate your diligence in ensuring the relevance and appropriateness of our visual content.

Comment 7. Some of the relevant literature should be reviewed and discussed, namely [1].

  • "State-Aware Anti-Drift Object Tracking," in IEEE Transactions on Image Processing, vol. 28, no. 8, pp. 4075-4086, Aug. 2019, doi: 10.1109/TIP.2019.2905984.

Response 7:

Thank you for recommending the inclusion of the specific literature. I have now reviewed and discussed the paper titled "State-Aware Anti-Drift Object Tracking," as published in IEEE Transactions on Image Processing, 2019. The discussion and citation can be found from line 90 on page 2 to line 93 on page 3. I appreciate your guidance in ensuring that our literature review is thorough.

Revisions: Building further on correlation filter enhancements, the State-aware Anti-drift Robust Correlation Tracking (SAT) by Han et al, introduces a method that integrates environmental context and a color-based reliability mask to refine correlation responses under internal disturbances.

Round 2

Reviewer 1 Report

Comments and Suggestions for Authors

No comment

Comments on the Quality of English Language

No

Reviewer 2 Report

Comments and Suggestions for Authors

this manuscript could be accepted